# Dimensioning Method of Floating Offshore Objects by Means of Quasi-Similarity Transformation with Reduced Tolerance Errors

**DOI:** 10.3390/s20226497

**Published:** 2020-11-13

**Authors:** Grzegorz Stępień, Arkadiusz Tomczak, Martin Loosaar, Tomasz Ziębka

**Affiliations:** 1Faculty of Navigation, Maritime University of Szczecin, 70-500 Szczecin, West Pomerania, Poland; a.tomczak@am.szczecin.pl; 2iSurvey, Westhill AB32 6FL, UK; martin.loosaar@gmail.com; 3Geometr Ltd., 71-525 Szczecin, Poland; tziebka@geometr.biz

**Keywords:** similarity transformation, affine transformation, rotation matrix, offshore surveying, dimensional control, orthogonal coordinate system, close range photogrammetry, total station

## Abstract

The human activities in the offshore oil and gas, renewable energy and construction industry require reliable data acquired by different types of hydrographic sensors: DGNSS (Differential Global Navigation Satellite System) positioning, attitude sensors, multibeam sonars, lidars or total stations installed on the offshore vessel, drones or platforms. Each component or sensor that produces information, unique to its position, will have a point that is considered as the reference point of that sensor. The accurate measurement of the offsets is vital to establish the mathematical relation between sensor and vessel common reference point in order to achieve sufficient accuracy of the survey data. If possible, the vessel will be put on a hard stand so that it can be very accurately measured using the standard land survey technique. However, due to the complex environment and sensors being mobilized when the vessel is in service, this may not be possible, and the offsets will have to be measured in sea dynamic conditions by means of a total station from a floating platform. This article presents the method of transformation by similarity with elements of affine transformation, called Q-ST (Quasi-Similarity Transformation). The Q-ST has been designed for measurements on such unstable substrates when it is not possible to level the total station (when the number of adjustment points is small (4–6 points)). Such situation occurs, among others, when measuring before the offshore duties or during the jack up or semi-submersible rig move. The presented calculation model is characterized by zero deviations at the adjustment points (at four common points). The transformation concerns the conversion of points between two orthogonal and inclined reference frames. The method enables the independent calculation of the scale factor, rotation matrix and system translation. Scaling is performed first in real space, and then both systems are shifted to the centroid, which is the center of gravity. The center of gravity is determined for the fit points that meet the criterion of stability of the orthogonal transformation. Then, the rotation matrix is computed, and a translation is performed from the computational (centroid) to real space. In the applied approach, the transformation parameters, scaling, rotation and translation, are determined independently, and the least squares method is applied independently at each stage of the calculations. The method has been verified in laboratory conditions as well as in real conditions. The results were compared to other known methods of coordinate transformation. The proposed approach is a development of the idea of transformation by similarity based on centroids.

## 1. Introduction

Similarity transformation is commonly used to convert coordinates between two three-dimensional orthogonal frames of reference [1]. This applies in particular to fields such as hydrography, geodesy, photogrammetry, offshore measurements and navigation [2,3,4]. Spatial transformations are commonly used in the processing of data from unmanned aerial vehicles [5], point clouds from laser scanning [6,7] and also with the use of computer vision algorithms [8]. Calculations of this type are also performed in engineering geodesy, where measurements are often converted to the coordinate system associated with the measured object, as well as when using, e.g., tunnel boring machines [9,10], and in many others. For big datasets of measurement, algorithms based on machine learning or applying the PCA (principal component analysis) method are also used for data processing [11,12,13].

Conformal transformation, also called transformation by similarity (or Helmert transformation), includes the change of scale (single accuracy) and isometry, which is related in the orthogonal transformation with translation and rotation [1,14,15]. This transformation concerns three groups of transformations: rotation, translation and scaling, and has seven parameters. The unknowns are: the translation vector, the scale factor and three rotation angles [16,17]. This transformation is usually presented in the form [18]:(1)$$\left[\begin{array}{c} X^{\prime }\\ Y^{\prime }\\ Z^{\prime } \end{array}\right]=\lambda \;\times \;R\;\times \;\left[\begin{array}{c} X\\ Y\\ Z \end{array}\right]+\left[\begin{array}{c} {\bar{X}}_{0}\\ {\bar{Y}}_{0}\\ {\bar{Z}}_{0} \end{array}\right]$$
where:${\left[{\bar{X}}_{0},\;{\bar{Y}}_{0},\;{\bar{Z}}_{0}\right]}^{T}$—translation vector,${\left[X,\;Y,\;Z\right]}^{T}$—vector (point) in the original system (subject to transformation),${\left[X^{\prime },\;Y^{\prime },\;Z^{\prime }\right]}^{T}$—vector (point) in the secondary system (treated as stationary),λ—scale factor,*R*—rotation matrix.

The rotation matrix *R* in Equation (1) is also commonly used in robotics, mechanics and automation and can be represented using various angle systems: Euler angles Equation (2), directional cosines Equation (3), angles used in photogrammetry Equation (4), angles related to motion, also referred to as Euler (or Tait-Bryan) angles Equation (5), as well as using Hamilton quaternions Equation (6) [18,19,20,21,22]. A rotation matrix using three angles of rotation can be represented by twelve sequences when all rotations are in the same direction (e.g., clockwise). Each sequence can be presented, depending on the adopted direction of the rotation, clockwise or anticlockwise, in six ways. This gives a total of seventy-two possibilities to build the rotation matrix. The in-depth analysis and the relationship between the various systems of angles and sequences of rotations and their relationships with Hamilton quaternions have been discussed in References [2,20,23,24]. In Figure 1 and in Equation (2), one of the possibilities of rotating with the use of the classic Euler angles system is presented, where rotation around one axis occurs twice (the order of rotations around the axis: *Z*—rotation by the angle *ψ*, *X* (*W*)—rotation by angle *θ*, *Z* (3)—rotation by angle *φ*).
(2)$$\begin{array}{cc} R & =\left[\begin{array}{ccc} \cos\psi & \sin\psi & 0\\ -\sin\psi & \cos\psi & 0\\ 0 & 0 & 1 \end{array}\right]\;\times \;\left[\begin{array}{ccc} 1 & 0 & 0\\ 0 & \cos\theta & \sin\theta \\ 0 & -\sin\theta & \cos\theta \end{array}\right]\;\times \;\left[\begin{array}{ccc} \cos\varphi & \sin\varphi & 0\\ -\sin\varphi & \cos\varphi & 0\\ 0 & 0 & 1 \end{array}\right]\\ & =\left[\begin{array}{ccc} -\cos\theta \times \sin\varphi \times \sin\psi +\cos\varphi \times \cos\psi & \cos\theta \times \cos\varphi \times \sin\psi +\cos\psi \times \sin\varphi & \sin\theta \times \sin\psi \\ -\cos\varphi \times \sin\psi -\cos\theta \times \cos\psi \times \sin\varphi & -\sin\varphi \times \sin\psi +\cos\theta \times \cos\varphi \times \cos\psi & \cos\psi \times \sin\theta \\ \sin\theta \times \sin\varphi & -\cos\varphi \times \sin\theta & \cos\theta \end{array}\right] \end{array}$$
where:rotation order: *ψ* → *θ* → *φ*, all rotations were assumed clockwise and with markings as in Figure 1,*ψ*—precession angle, between the OX axis and the OW node line (rotation around the Z axis),*θ*—nutation angle, between the OZ and O3 axes (rotation around the OW node line),*φ*—angle of pure rotation (intrinsic rotation), between the line of nodes OW and the axis O1 (rotation around axis 3).

The rotation matrix is also represented by directional cosines, where individual columns of the R matrix are represented by the scalar products of the dissipers of individual systems. The target system is also referred to in the literature as a stationary or reference frame [2,20]:(3)$$R=\left[\begin{array}{ccc} \left.cos(x,x^{\prime }\right) & \left.cos(x,y^{\prime }\right) & \left.cos(x,z^{\prime }\right)\\ \left.cos(y,x^{\prime }\right) & \left.cos(y,y^{\prime }\right) & \left.cos(y,z^{\prime }\right)\\ \left.cos(z,x^{\prime }\right) & \left.cos(z,y^{\prime }\right) & \left.cos(z,z^{\prime }\right) \end{array}\right]$$

The rotation matrix in photogrammetry is determined on the basis of the angles related to the motion in the following order: *ω* (roll), *φ* (pitch), *κ* (yaw), which is written in the form of Equation (4):(4)$$\begin{array}{cc} R & =\left[\begin{array}{ccc} 1 & 0 & 0\\ 0 & \cos\omega & \sin\omega \\ 0 & -\sin\omega & \cos\omega \end{array}\right]\times \left[\begin{array}{ccc} \cos\varphi & 0 & -\sin\varphi \\ 0 & 1 & 0\\ \sin\varphi & 0 & \cos\varphi \end{array}\right]\times \left[\begin{array}{ccc} \cos\kappa & \sin\kappa & 0\\ -\sin\kappa & \cos\kappa & 0\\ 0 & 0 & 1 \end{array}\right]\\ & =\left[\begin{array}{ccc} cos\varphi \times cos\kappa & cos\varphi \times sin\kappa & -sin\varphi \\ -cos\omega \times sin\kappa +sin\omega \times sin\varphi \times cos\kappa & cos\omega \times cos\kappa +sin\omega \times sin\varphi \times sin\kappa & sin\omega \times cos\varphi \\ sin\omega \times sin\kappa +cos\omega \times sin\varphi \times cos\kappa & -sin\omega \times cos\kappa +cos\omega \times sin\varphi \times \sin\kappa & cos\omega \times cos\varphi \end{array}\right] \end{array}$$
where:rotation order: *ω* → *φ* → *κ*,*ω*—the rotation around the *X* axis (clockwise), the roll angle,*φ*—the rotation around the *Y* axis (anticlockwise), the pitch angle,*κ*—the rotation around the *Z* axis (clockwise), the yaw angle.

In navigation, extended notation of the basic Euler system of angles is used and they are often defined in relation to the body frame to an external (often defined by GNSS – Global Navigation Satellite System) reference frame (fixed frame). The rotation sequence is then the opposite to that described with the Equation (4), which for an orthogonal transformation can be written as a transposition of this matrix in the form of the Equation (5):(5)$$\begin{array}{cc} R & =\left[\begin{array}{ccc} \cos\psi & -\sin\psi & 0\\ \sin\psi & \cos\psi & 0\\ 0 & 0 & 1 \end{array}\right]\times \left[\begin{array}{ccc} \cos\theta & 0 & \sin\theta \\ 0 & 1 & 0\\ -\sin\theta & 0 & \cos\theta \end{array}\right]\times \left[\begin{array}{ccc} 1 & 0 & 0\\ 0 & \cos\varphi & -\sin\varphi \\ 0 & \sin\varphi & \cos\varphi \end{array}\right]\\ & =\left[\begin{array}{ccc} cos\theta \times cos\psi & -cos\varphi \times sin\psi +sin\varphi \times sin\theta \times cos\psi & sin\varphi \times sin\psi +cos\varphi \times sin\theta \times cos\psi \\ cos\theta \times sin\psi & cos\varphi \times cos\psi +sin\varphi \times sin\theta \times sin\psi & -sin\varphi \times cos\psi +cos\varphi \times sin\theta \times \sin\psi \\ -sin\theta & sin\varphi \times cos\theta & cos\varphi \times cos\theta \end{array}\right] \end{array}$$
where:rotation order: *ψ* → *θ* → *φ*,*ψ*—yaw angle (anticlockwise), rotation around the *Z* axis,*θ*—pitch angle (clockwise), rotation around the *Y* axis (after rotation around the *Z* axis),*φ*—roll angle (anticlockwise), rotation around the *X* axis (after rotation around the *Z* and *Y* axes).

The rotation matrix (5) written with Hamilton quaternions will take the form of the relationship (6). The relation between the Euler angles assumed in Equation (5) and the quaternions described with Equation (6) will then take the form of Equation (7), with the norm in the form of Equation (8):(6)$$Q=\left[\begin{array}{ccc} q_{1}{}^{2}+q_{2}{}^{2}-q_{3}{}^{2}-q_{4}{}^{2} & 2\left(q_{2}\times q_{3}-q_{1}\times q_{4}\right) & 2\left(q_{2}\times q_{4}+q_{1}\times q_{3}\right)\\ 2\left(q_{2}\times q_{3}+q_{1}\times q_{4}\right) & q_{1}{}^{2}-q_{2}{}^{2}+q_{3}{}^{2}-q_{4}{}^{2} & 2\left(q_{3}\times q_{4}-q_{1}\times q_{2}\right)\\ 2\left(q_{2}\times q_{4}-q_{1}\times q_{3}\right) & 2\left(q_{3}\times q_{4}+q_{1}\times q_{2}\right) & q_{1}{}^{2}-q_{2}{}^{2}-q_{3}{}^{2}-q_{4}{}^{2} \end{array}\right]$$
where:(7)$$q_{1}=\cos\frac{\varphi }{2}\;\times \;\cos\frac{\theta }{2}\times \cos\frac{\psi }{2}+\sin\frac{\varphi }{2}\times \sin\frac{\theta }{2}\times \sin\frac{\psi }{2}\;q_{2}=\sin\frac{\varphi }{2}\;\times \cos\frac{\theta }{2}\times \cos\frac{\psi }{2}-\cos\frac{\varphi }{2}\times \sin\frac{\theta }{2}\times \sin\frac{\psi }{2}\;q_{3}=\cos\frac{\varphi }{2}\;\times \sin\frac{\theta }{2}\times \cos\frac{\psi }{2}+\sin\frac{\varphi }{2}\times \cos\frac{\theta }{2}\times \sin\frac{\psi }{2}\;q_{4}=\cos\frac{\varphi }{2}\;\times \cos\frac{\theta }{2}\times \sin\frac{\psi }{2}-\sin\frac{\varphi }{2}\times \sin\frac{\theta }{2}\times \cos\frac{\psi }{2}$$
(8)$$q_{1}{}^{2}+q_{2}{}^{2}+q_{3}{}^{2}+q_{4}{}^{2}=1$$

Equations (6)–(8), although not applied in the practical part of the study, are presented in addition to the completeness of the theoretical considerations.

The rotation matrix is also presented in a parametric Equation (9), which is very convenient to use because it can then be used to calculate the rotation matrix in any system of angles:(9)$$M=\left[\begin{array}{ccc} m_{11} & m_{12} & m_{13}\\ m_{21} & m_{22} & m_{23}\\ m_{31} & m_{32} & m_{33} \end{array}\right]$$

In general, the transformation matrix M has nine unknowns, including six independent ones, but after applying the orthogonal transformation Equation (10), only three of them will be linearly independent, which leads to Equation (11):(10)$$m_{11}^{2}+m_{12}^{2}+m_{13}^{2}=1\;m_{21}^{2}+m_{22}^{2}+m_{23}^{2}=1\;m_{31}^{2}+m_{32}^{2}+m_{33}^{2}=1\;m_{11}\times m_{12}+m_{21}\times m_{22}+m_{31}\times m_{32}=0\;m_{11}\times m_{13}+m_{21}\times m_{23}+m_{31}\times m_{33}=0\;m_{12}\times m_{13}+m_{22}\times m_{23}+m_{32}\times m_{33}=0$$
(11)$$M\times M^{T}=I$$
where:*I*—identity matrix.

In the case of the affine transformation, in the nine-unknown approach, the scale factor is replaced by the scale factor vector $\lambda =diag{\left[\lambda_{X},\;\lambda_{Y},\;\lambda_{Z}\right]}^{T}$, which allows a different scale for each axis. If all the scale factors are the same and after the orthogonality condition in the Equation (10) or Equation (11) is satisfied by the rotation matrix, the affine transformation goes into a transformation by similarity. Various variants of the affine transformation are presented, e.g., in References [25,26,27], and in this study, they will not be described in more detail.

In practice, in many applications, the rotation matrix is reduced to the infinitesimal (aiming at zero) values of the angles and presented in the form of the Equation (12):(12)$$m=\left[\begin{array}{ccc} 1 & -\psi & \theta \\ \psi & 1 & -\varphi \\ -\theta & \varphi & 1 \end{array}\right]$$
where:*ψ*, *θ*, *φ*—infinitesimal values of angles expressed in radians, the angles are the same as in Equation (5).

The matrix (12) is also called the small rotation angles due to the infinitesimal values of the angles (expressed in radians).

Transformation by similarity based on infinitesimal values of angles (expressed in radians), with a lattice scale close to one, will take the form described as the Buršy–Wolf transformation [28,29]:(13)$$\left[\begin{array}{c} X\\ Y\\ Z \end{array}\right]=\left(1+ds\right)\times \left[-\begin{array}{ccc} 1 & \epsilon_{Z} & -\epsilon_{Y}\\ \epsilon_{Z} & 1 & \epsilon_{X}\\ \epsilon_{Y} & -\epsilon_{X} & 1 \end{array}\right]\;\times \left[\begin{array}{c} X^{\prime }\\ Y^{\prime }\\ Z^{\prime } \end{array}\right]+\left[\begin{array}{c} t_{X}\\ t_{Y}\\ t_{Z} \end{array}\right]$$
where:$\epsilon_{Z}$—rotation angle around the *Z* axis,$\epsilon_{Y}$—rotation angle around the *Y* axis,$\epsilon_{X}$—rotation angle around the *X* axis,1+ds—scale factor, where *ds* corresponds to linear system distortion,${\left[t_{X},\;t_{Y},\;t_{Z}\right]}^{T}$—translation vector.

The rotation matrix expressed in Equation (13) is the inverse of the matrix shown in Equation (12).

On the other hand, Badekas [30] provides a record of the transformation between the average terrestrial system and the geodetic system in the form of the relationship (14):(14)$$\left[\begin{array}{c} X\\ Y\\ Z \end{array}\right]=\left[\begin{array}{c} d_{x_{0}}\\ d_{y_{0}}\\ d_{z_{0}} \end{array}\right]+\left[\begin{array}{c} X_{0}\\ Y_{0}\\ Z_{0} \end{array}\right]+\left[\begin{array}{ccc} 1 & da_{3} & -da_{2}\\ -da_{3} & 1 & da_{1}\\ da_{2} & -da_{1} & 1 \end{array}\right]\times \left[\begin{array}{c} x-x_{0}\\ y-y_{0}\\ z-z_{0} \end{array}\right]+\epsilon \left[\begin{array}{c} x-x_{0}\\ y-y_{0}\\ z-z_{0} \end{array}\right]$$
where:
${\left[X\;Y\;Z\right]}^{T}$—coordinates in the average Earth coordinate system,${\left[x\;y\;z\right]}^{T}$—coordinates in a geodetic (local) coordinate system,$da_{1},\;da_{2},\;da_{3}$—(infinitesimal) rotation angles expressed in radians,ϵ—scale correction,${\left[\begin{array}{ccc} X_{0} & Y_{0} & Z_{0} \end{array}\right]}^{T}$—vector of translation in the terrestrial system,${\left[\begin{array}{ccc} x_{0} & y_{0} & z_{0} \end{array}\right]}^{T}$—translation vector in the geodetic (local) system,${\left[\begin{array}{ccc} d_{x_{0}} & d_{y_{0}} & d_{z_{0}} \end{array}\right]}^{T}$—the coordinates of the origin of the geodetic (local) coordinate system after rotation and shift to the mean terrestrial system.

In the transformation described by Equation (14), both the reference frame and the geodetic (local) system are shifted to a common centroid, which is the center of gravity of the Earth system. The transformation described by Equation (14) written in accordance with the Buršy–Wolf transformation (13) functions in the literature under the name of the Molodensky−Badekas transformation (15), although many authors present it in different variants [29]:(15)$${\left[\begin{array}{c} X\\ Y\\ Z \end{array}\right]}_{2}=\left(1+ds\right)\left[-\begin{array}{ccc} 1 & \epsilon_{Z} & -\epsilon_{Y}\\ \epsilon_{Z} & 1 & \epsilon_{X}\\ \epsilon_{Y} & -\epsilon_{X} & 1 \end{array}\right]\times {\left[\begin{array}{c} \bar{X}\\ \bar{Y}\\ \bar{Z} \end{array}\right]}_{1}+{\left[\begin{array}{c} t_{X}\\ t_{Y}\\ t_{Z} \end{array}\right]}_{2}$$
where:
${\left[\begin{array}{ccc} \bar{X} & \bar{Y} & \bar{Z} \end{array}\right]}^{T}{}_{1}$—coordinates in the geodetic (local) system after moving to the centroid, the center of gravity of this system,indexes 1 and 2 next to the translation and coordinate vectors mean, respectively: 1—geodetic (local) system, 2—Earth system.

Thus, in Equation (15), there is a double translation, first, the geodetic system (local, marked as 1) is shifted to its own center of gravity, and after rotation and scale correction, a re-translation takes place—back to the Earth system, and inversely to the one to which the Earth system was subjected at the Buršy–Wolf and Molodensky−Badekas transformations can also be written using one of the rotation matrices presented in Equations (2)–(6) and (9) for large values of rotation angles. Centroid as the center of gravity and the influence on its location on the accuracy of coordinates calculation is presented in Reference [31].

Infinitesimal values of transformation parameters are used in the adjustment process, where the least squares method is most often used. The discussion on the use of the least squares method in conformal and affine transformations has been undertaken in many studies, including References [6,16,18,32,33,34]. The adjustment procedure leads to the search for the most probable solutions, which is associated with the minimization of errors [35,36] and directly derived from the maximization of the probability, *P*:
(16)$$P=\frac{1}{\sigma \sqrt{2\pi }}\int_{-t}^{t}e^{-\frac{{\left(x-\mu \right)}^{2}}{2\sigma }}dx$$

The solution (calculation of the vector of unknowns) is presented in a matrix form and written in the form:(17)$$X={\left(A^{T}\times P\times A\right)}^{-1}\times A^{T}\times P\times L$$
where:*X*—vector of unknown parameters,*A*—matrix of coefficients with unknowns (partial derivatives)—Jacobian transformation,*L*—vector of constants,*P*—vector of weighting factors (statistical weights).

In practice, the adjustment problem comes down to the formulation of the correction Equation (18) or the formulation of the computational problem in the Equation (19), where one of them is calculated, while another is provided by the data [6,32]:(18)V=A×X−L
(19)L=A×X

The correction Equation (18) for the transformation by similarity (1) with the use of a small-rotation matrix (12) from the body frame to the stationary frame (reference frame–fixed frame) can be written as Equation (20), and after transformation into the Equation (21) [37]:(20)$$\left[\begin{array}{c} X^{\prime }\\ Y^{\prime }\\ Z^{\prime } \end{array}\right]-{\left[\begin{array}{c} x\\ y\\ z \end{array}\right]}_{transformed}=\left[\begin{array}{ccc} d\lambda & -\psi & \theta \\ \psi & d\lambda & -\varphi \\ -\theta & \varphi & d\lambda \end{array}\right]\times \;\left[\begin{array}{c} x\\ y\\ z \end{array}\right]+\left[\begin{array}{c} dx_{0}\\ dy_{0}\\ dz_{0} \end{array}\right]$$
(21)$$\left[\begin{array}{c} V_{X}\\ V_{Y}\\ V_{Z} \end{array}\right]=\left[\begin{array}{ccccccc} x & 0 & -y & z & 1 & 0 & 0\\ y & -z & x & 0 & 0 & 1 & 0\\ z & y & 0 & -x & 0 & 0 & 1 \end{array}\right]\times \left[\begin{array}{c} d\lambda \\ \begin{array}{c} \varphi \\ \psi \\ \theta \end{array}\\ \begin{array}{c} dx_{0}\\ dy_{0}\\ dz_{0} \end{array} \end{array}\right]+\left[\begin{array}{c} X^{\prime }\\ Y^{\prime }\\ Z^{\prime } \end{array}\right]-{\left[\begin{array}{c} x\\ y\\ z \end{array}\right]}_{transformed}$$

The reference of compound (21) to the designations of compound (18) is as follows:(22)$$V=\left[\begin{array}{c} V_{X}\\ V_{Y}\\ V_{Z} \end{array}\right],\;A=\left[\begin{array}{ccccccc} x & 0 & -y & z & 1 & 0 & 0\\ y & -z & x & 0 & 0 & 1 & 0\\ z & y & 0 & -x & 0 & 0 & 1 \end{array}\right],\;X=\left[\begin{array}{c} d\lambda \\ \begin{array}{c} \varphi \\ \psi \\ \theta \end{array}\\ \begin{array}{c} dx_{0}\\ dy_{0}\\ dz_{0} \end{array} \end{array}\right],\;L=\left[\begin{array}{c} X^{\prime }\\ Y^{\prime }\\ Z^{\prime } \end{array}\right]-{\left[\begin{array}{c} x\\ y\\ z \end{array}\right]}_{transformed}$$

In this study, the least squares method is used in stages, and not in one equalization process, e.g., expressed by Equation (20). It is used separately at each stage of the transformation, and therefore it is abbreviated as S2-LSM (Stages Sequence–Least Square Method). Such an approach is implemented in the Quasi-Similarity Transformations (Q-ST) algorithm and results in resetting the deviations on the combined points for four adjustment points. This method was designed for measurements in dynamic inclined systems, in particular in the offshore industry and in close-range geodesy and photogrammetry.

The aim of this publication is to develop a coordinate transformation method dedicated to measurements in a dynamic body frame, ensuring simplicity and speed of calculations with a simultaneous reduction to the minimum of errors on the adjustment points, when their number is small (4–6 points). According to the authors, the developed Q-ST method can be used to convert coordinates in the offshore industry and in traditional geodetic and photogrammetric measurements. The method can also be used in navigation, automation, mechanics and robotics. The advantage of the proposed method is especially visible in the case of a small number of adjustment points (4 or 5), and by zeroing the errors at the adjustment points, according to Equation (16)—the calculated coordinates can be treated as maximally probable.

## 2. Materials and Methods 

### 2.1. Vessel Offsets Measurements

To achieve the highest accuracy standards, vessels’ offsets determination is usually performed with total station. Total station, as in other industries, is an essential instrument used in the offshore industry for surveying and building construction due to the level of accuracy it provides. They are used for dimensional control of offshore vessels, survey boats, jack up and semi-submersible platforms, wind turbines, remotely operated vehicles, including primarily the measurement of offsets of navigational or hydrographic sensors and measurements, and other elements of ship’s structure, essential for offshore operations.

Precisely measured offsets in the local (vessel) coordinate frame are entered into the hydrographic software integrating signals from sensors, e.g., GNSS, Multi-beam echosounder, gyrocompass, motion reference unit, acoustic underwater positioning systems, lidars and others, and based on the adopted navigation solution, e.g., Kalman filter, acquired data X, Y, Z are logged in the global coordinate frame.

The robotic total station is also used as a navigation sensor for positioning of the jack up and semi-submersible platforms during approach to installation of offshore structures, e.g., during jacket or wind turbine pile installation projects. They are an alternative to GNSS satellite positioning wherever there is a risk of signal obstruction and loss of platform position. Based on both vertical and horizontal angles and the slope distance, the navigation solution is performed. Total Station mounted on the vessel takes measurements to a particular point (usually prism reflector installed on the offshore installation with known coordinates) and the dynamic vessel’s position is determined by hydrographic software and can be monitored to mitigate the risk of collision. 

During the survey, the tilt compensator is locked. Each survey station is established in a random (local) coordinate system on board the floating vessel. Usually more than one survey station is needed for covering all the observations. Therefore, at least four common control points between survey stations are established and observed. Additionally, for measuring the sensors of interest, the vessel body must be surveyed for center line alignment and the positioning of the new established local coordinate frame of the vessel (see Figure 2).

As already mentioned, all the survey stations are independent and established in a random coordinate system. The instrument does not need to be levelled, but for avoiding any bending of the instrument under its own weight, it is good practice to level it approximately. The tilt compensator must be locked, because the instrument must be fixed relatively to the vessel when the vessel is floating. As a precaution for discovering small accidental moves of the instrument during the survey (hitting the tripod with leg, for example), it is wise to start the survey station with measuring the farthest point and to finish the survey station with re-measuring that point. By comparing the initial coordinates against the last ones, a possible instrument move becomes visible. Each survey station must be connected to others through at least four control points. Even though three are minimum, it does not give enough quality control. Good practice is having five or more common points between stations. 

Planning and preparing the survey is the most time-consuming part of the whole process. It includes establishing the control points at suitable places all over the vessel (see Figure 3) and planning the survey station locations. Because usually there are multiple operations simultaneously progressing on board the vessel, the surveyor must prepare for sudden unexpected visibility issues raised during the observations. Also, the geometry of the common control points between survey stations must be wide enough for providing wanted precision (due to our practice). A proven approach is: starting the survey from the location with the widest visibility and measuring many control points on all parts of the vessel from there. Other survey stations established later are referenced to the points from the first station. This way, the “chain of stations” is as short as practically possible. In the best-case scenario, all the stations are connected directly to the first survey station control points.

The meaning of wide geometry of common control points between stations is different from the meaning land surveyors are used to. On terrestrial observations, the *Z* axis is locked with the help of gravity (by levelling the instrument) and the survey can be considered as two-dimensional regarding the rotations. It is enough to measure 2 points with decent distance between them for establishing the position of the total station. However, on a floating object, at least 3 points are needed for establishing the position of the instrument, and these 3 points cannot be adjusted with LSM. In a perfect case, they form an equilateral triangle. While planning the survey, the surveyor must analyze the situation in three-dimension (3D)—control points should stretch out to various directions.

Usually, some navigation systems on board ships already have a local coordinate system defined, and for simplicity, the surveyor should use the same location for his coordinate frame origin as specified in vessel systems (Ultra Short Baseline (USBL) transducer in Dynamic Positioning system (DP)). That means, during the total station observations, a common element must be surveyed, for knowing where the pre-defined coordinate frame’s origin is located. That element is an observable sensor, which coordinates (*X*, *Y*, *Z* values) can be read out from the navigation system, for example, USBL pole or GPS antenna. The origin of the coordinate system of navigation systems can therefore be determined during surveys in the local random coordinate system of the total station.

Below can be found the complete list of items to observe during the total station survey (in addition to the control points for the current survey and for future use—marked in Figure 3):
Sensors of interest (the purpose of the survey),Sensors for connecting the survey to a pre-established coordinate frame of vessel system (USBL, DP),Ship elements for connecting the survey to General Arrangement (GA) drawing of a vessel,Prepared pitch, roll and heading calibration points,Ship body for alignment and positioning of the new local coordinate frame:*Z*-axis direction (base plane),*Y*-axis direction,Origin *X*-position (center line of vessel),Origin *Y*-position,Origin *Z*-position.

Data processing is done in dimensional control software, like, for example, SC4W. For simplicity, the survey data is imported into the software as (*X*,*Y*,*Z*) coordinates in 1.0 or 0.1 mm precision, instead of raw angles and distances. The next step is merging the survey stations. The first survey station is considered as fixed and other survey stations are rotated and shifted in the best possible way to match common control points with the first station. This is usually done with the help of data processing software’s least squares best fit function. After the merging process, all the surveyed points are in the form of one point cloud instead of independent unaligned point clouds of each station. During the data processing, all survey stations are merged, and one point cloud is formed. After that, the point cloud is aligned with the vessel body and as the effect, the origin of the coordinate frame is positioned on the desired location. After the aligning process and coordinate frame positioning related to the vessel is completed, the sensor coordinates are obtained.

For quality control and for positioning the surveyed points relative to the vessel, it is useful to have GA drawings of the ship in CAD format. For being able to relate the GA drawing with total station survey, few well-defined common elements must be found which are easily observable and also presented on the GA drawing. In most cases, vessel attitude sensor calibration follows the offsets survey. Therefore, it is wise to pre-install poles for RTK (Real Time Kinematic) Rovers or prisms, depending on the calibration method. These poles are observed during the total station survey and local coordinates are obtained.

Due to the presented approach, similarity transformation algorithms using the least squares method are used to connect point clouds. For on-board measurements, as mentioned, the number of common points is generally four or five. This publication presents a similarity transformation method that is suited to this small number of common points. This method is characterized by a minimum value of errors of alignment at common points and is based on the Q-ST algorithm.

### 2.2. Quasi-Similarity Transformations (Q-ST) Algorithm

In the presented Quasi-Similarity Transformations (Q-ST) method, it was assumed that the transformed system associated with the measurement instrument is named as local frame—LF, and the target—stationary system to which the transformation takes place, was named vessel frame—VF (fixed-frame). The composition of the transformation groups: rotations, translations and scaling, is arbitrary in the transformation and results in a mathematical notation appropriate to the adopted model. The functional model of the Q-ST transformation is presented by the formula (23):(23)$$X_{VF}=R\times \left(\lambda \cdot X_{LF}-X_{CLF}\right)+X_{CVF}$$
where:
$X_{VF}$—coordinates in the target system—stationary (vessel reference frame—VF),R—rotation matrix,λ—scale factor,$X_{LF}$—coordinates in the transformed system (local reference frame—LF),$X_{CLF}$—coordinates of the center of gravity in the transformed system (centroid of local reference frame),$X_{CVF}$—coordinates of the center of gravity in the target system—stationary (centroid of vessel reference frame).

The Q-ST (23) is performed in the steps shown in Figure 4.

After collecting the data by means of measurements in the local system of the LF instrument (local reference frame), they are converted to the vessel’s VR coordinate system (vessel reference frame). The calculations are performed in the steps shown in Figure 4. Below are descriptions of each step of the Q-ST.

Stage 1:

The first stage of transformation is related to the scale adjustment between the LF and VF systems. The scale factor λ for a pair of homologous vectors—built from points known in both systems—LF and VF—is determined from the dependence (24):(24)$$D_{VF}=\lambda \times D_{LF}$$
where:
$D_{VF}$—vector length in the ship’s coordinate system (VF),$D_{LF}$—the length of the vector in the local coordinate system associated with the instrument (LF).

The scale factor, using Equation (24), can be determined independently for each pair of homologous vectors known in the VF and LF systems. Calculation of the global scale factor between the LF and VF systems for all homologous vectors is performed using the least squares method. The computational problem is reduced to the form of the relationship (19), and the solution written according to the Formula (17) will then take the form of the relationship (25):(25)$$\lambda ={\left(D_{LF}{}^{T}\times D_{LF}\right)}^{-1}\times D_{LF}{}^{T}\times D_{VF}$$
where:
$D_{LF}={\left[D_{1},D_{2},\ldots ,D_{N}\right]}^{T}$—a vector consisting of the length of the homologous vectors in the local LF system,$D_{VF}={\left[D_{1},D_{2},\ldots ,D_{N}\right]}^{T}$—a vector consisting of the lengths of the homologous vectors in the VF ship system,$D_{1},D_{2},\ldots ,D_{N}$—lengths of homologous vectors in the LF and VF coordinate systems,N—the number of homolog vector pairs known in LF and VR.

The reference of the signs in Equation (25) to the signs in Equations (17) and (19) is as follows:(26)$$A=D_{LF},\;X=\lambda ,\;L=D_{VF}$$

At this stage of transformation, the homogeneity of scale changes is also tested, which is related to the behavior of the orthogonal transformation criterion (robust orthogonal criterion in Figure 4). For each pair of homologous vectors in LF and VF, scale corrections are made, which according to relation (18) and notations (26) and taking into account the Equation (25), can be written as Equation (27):(27)$$V=D_{LF}\times {\left(D_{LF}{}^{T}\times D_{LF}\right)}^{-1}\times D_{LF}{}^{T}\times D_{VF}-D_{VF}$$

For the corrections calculated with the use of the Equation (27), the standard deviation $\sigma_{D}$ is calculated using Equation (28):(28)$$\sigma_{D}=\sqrt{\frac{V^{T}\times V}{N-1}}$$

Then, for each deviation from the calculation scale, the condition written in the form (29) is checked using the Equation (27):(29)$$V_{i}\geq 2\sigma_{D}$$

The deviations from the scale testify to the heterogeneity of its change between the LF and VF systems. The criterion of exceeding the double value of the standard deviation is statistically justified for twenty lengths of homologous vectors, in the case of normal distribution of their values. In the case of five connecting points known in both systems, ten pairs of homologous vectors can be constructed on their basis. Therefore, exceeding the standard deviation twice can be treated as a disturbance of the scale, which in this case can also be justified using the Chauvenet criterion [36]. Condition (29) is a notation of a robust orthogonality transformation criterion. According to this condition, pairs of homologous vectors for which the condition (29) is satisfied introduce a scale disturbance and are excluded from the determination of this coefficient. After their elimination, the scale factor is determined again using the Formula (25).

Stage 2:

In the second step, the center of gravity of both the LF and VF coordinate systems is determined. The coordinates of the center of gravity have the form of relations (32) and are the result of searching for the extreme (minimum) of the function *F* (30), for which the sum of the squares of the distance from the searched point is the minimum:(30)$$F\left(x,y,z\right)=D_{1}{}^{2}+D_{2}{}^{2}+\ldots +D_{n}{}^{2}\;D_{i}{}^{2}={\left(x_{C}-x_{i}\right)}^{2}+{\left(y_{C}-y_{i}\right)}^{2}+{\left(z_{C}-z_{i}\right)}^{2}=minimum$$

At the stationary point, the partial derivatives of the function (30) take the extreme (minimum) value, which was written using the compounds (31). Directly from the compounds (31), after ordering, the coordinates of the center of gravity are obtained in the form (32) [38]. The center of gravity is therefore determined by minimizing the squared distances to each of the common points in the LF and VF systems:(31)$$\frac{\partial F}{\partial x}=2\left(x_{C}-x_{1}\right)+2\left(x_{C}-x_{2}\right)+\ldots +2\left(x_{C}-x_{n}\right)=2\left(nx_{C}-x_{1}-x_{2}-\ldots -x_{n}\right)=0\;\frac{\partial F}{\partial y}=2\left(y_{C}-y_{1}\right)+2\left(y_{C}-y_{2}\right)+\ldots +2\left(y_{C}-y_{n}\right)=2\left(ny_{C}-y_{1}-y_{2}-\ldots -y_{n}\right)=0\;\frac{\partial F}{\partial z}=2\left(z_{C}-z_{1}\right)+2\left(z_{C}-z_{2}\right)+\ldots +2\left(z_{C}-z_{n}\right)=2\left(nz_{C}-z_{1}-z_{2}-\ldots -z_{n}\right)=0$$
where:
$D_{i}$—length of the homologous vector in each of the LF and VF systems built using two points: the center of gravity and the considered fit point,$X_{C},\;Y_{C},Z_{C}$—coordinates of the center of gravity (center of mass) in LF and VF respectively,$X_{i},\;Y_{i},Z_{i}$—the coordinates of the fitting point, i ∈ (1,2, ..., *n*),n—number of homologous points, adjustments (common points) in LF and RF systems.

(32)$$X_{CLF}=\frac{\sum_{i=1}^{N}\left(X_{i}\right)}{n},\;Y_{CLF}=\frac{\sum_{i=1}^{N}\left(Y_{i}\right)}{n},\;Z_{CLF}=\frac{\sum_{i=1}^{N}\left(Z_{i}\right)}{n},\;X_{CVF}=\frac{\sum_{i=1}^{N}\left(X_{i}\right)}{n},\;Y_{CVF}=\frac{\sum_{i=1}^{N}\left(Y_{i}\right)}{n},\;Z_{CVF}=\frac{\sum_{i=1}^{N}\left(Z_{i}\right)}{n},$$

Stage 3:

In the third stage, auxiliary coordinates are determined in the LF and VF systems. These coordinates are calculated for the common points after translating both systems into the centroid—the center of gravity of both systems. The idea of translation is presented in Figure 5. 

The systems (Figure 5) are translated along the axis of each of the LF and VF coordinate systems. In this way, the mutual angular orientation of the two systems does not change. As a result, new centroid coordinates are obtained, in the auxiliary computational space, and related to the center of gravity. These coordinates for points in LF and VF systems, after taking into account relations (25) and (32), can be written using Formula (33):(33)$$X_{iLF}=\left(\lambda \times X_{LF}-X_{CLF}\right)=\left[{\left(D_{LF}{}^{T}\times D_{LF}\right)}^{-1}\times D_{LF}{}^{T}\times D_{VF}\right]\times X_{LF}-\frac{\sum_{i=1}^{N}\left(X_{i}\right)}{n}\;X_{iVF}=\left(X_{VF}-X_{CVF}\right)=X_{VF}-\frac{\sum_{i=1}^{N}\left(X_{i}\right)}{n}$$
where:
$X_{iLF}$—centroid coordinates of points in the LF and VF systems related to the center of gravity,$X_{iVF}$—centroid coordinates of points in the VF system.

Stage 4:

In the fourth step, the rotation matrix of the system LF with respect to VF is determined. The matrix of rotation is calculated using the coordinates defined by Equation (33). The designations of the angles of the LF system with respect to VF were adopted in accordance with the relationship (5) and Figure 5. For the simplicity and convenience of calculations, a parametric matrix (9) is used. For three common points, the rotation matrix is determined from the dependence (34):(34)$$M=X_{iVF}\times X_{iLF}{}^{-1}$$
where:
M—a rotation matrix defined by Equation (9),$X_{iLF}$, $X_{iVF}$—designations as in Equation (33).

In the case when the number of adjustment points is greater than three, the rotation matrix *M* is determined by the least squares method according to the relationship (17), and the Equation (34) is redefined to the form (19), which can be written in the form of the relationship (35):(35)$$[\begin{array}{c} X_{iVF1}\\ Y_{iVF1}\\ Z_{iVF1}\\ X_{iVF2}\\ Y_{iVF2}\\ Z_{iVF2}\\ \ldots \\ X_{iVFN}\\ X_{iVFN}\\ X_{iVFN} \end{array}]=\left[\begin{array}{ccccccccc} X_{iLF1} & Y_{iLF1} & Z_{iLF1} & 0 & 0 & 0 & 0 & 0 & 0\\ 0 & 0 & 0 & X_{iLF1} & Y_{iLF1} & Z_{iLF1} & 0 & 0 & 0\\ 0 & 0 & 0 & 0 & 0 & 0 & X_{iLF1} & Y_{iLF1} & Z_{iLF1}\\ X_{iLF2} & Y_{iLF2} & Z_{iLF2} & 0 & 0 & 0 & 0 & 0 & 0\\ 0 & 0 & 0 & X_{iLF2} & Y_{iLF2} & Z_{iLF2} & 0 & 0 & 0\\ 0 & 0 & 0 & 0 & 0 & 0 & X_{iLF2} & Y_{iLF2} & Z_{iLF2}\\ X_{iLFN} & Y_{iLFN} & Z_{iLFN} & 0 & 0 & 0 & 0 & 0 & 0\\ 0 & 0 & 0 & X_{iLFN} & Y_{iLFN} & Z_{iLFN} & 0 & 0 & 0\\ 0 & 0 & 0 & 0 & 0 & 0 & X_{iLFN} & Y_{iLFN} & Z_{iLFN} \end{array}\right]\times [\begin{array}{c} m_{11}\\ m_{12}\\ m_{13}\\ m_{21}\\ m_{22}\\ m_{23}\\ m_{31}\\ m_{32}\\ m_{33} \end{array}]$$
where
$$X={\left(A^{T}\times A\right)}^{-1}\times A^{T}\times L$$

The vector of unknowns, *X*, calculated using the Equation (35), contains elements of the rotation matrix *M*. The matrix *M* describes the relation between the centroid coordinates in the instrument (LF) and ship (VF) systems, which are given by the Equation (33). This relationship can be written in a parametric form in Equation (36):(36)$$X_{iVF}=M\times X_{iLF}$$

Equation (36) has a simple form, but it does not guarantee that the elements of the matrix *M* are related only to rotations, which can be stated after checking the orthogonality condition in the form (11). In the case of an orthogonal transformation, the condition (11) should be strictly met, which for a coordinate transformation containing six significant digits in the discussed case means compliance with the unit matrix at the level of $1\times 10^{-6}$. Such a match confirms the condition of orthogonality and transformation by similarity. Otherwise, we should look at the *M* matrix as a transformation matrix—a quasi-rotation matrix—which we denote here as *QR*. The *QR* matrix is a 3 × 3 matrix and it can also be obtained by the affine transformation of coordinates by Equation (36), which can be written in the form of Equations (37) and (38):(37)$$X_{iVF}=\mathrm{QR}\times X_{iLF}$$
where:(38)$$QR=\left[\begin{array}{ccc} \lambda_{X} & 0 & 0\\ 0 & \lambda_{Y} & 0\\ 0 & 0 & \lambda_{Z} \end{array}\right]\times \left[\begin{array}{ccc} m_{11} & m_{12} & m_{13}\\ m_{21} & m_{22} & m_{23}\\ m_{31} & m_{32} & m_{33} \end{array}\right]=\left[\begin{array}{ccc} q_{11} & q_{12} & q_{13}\\ q_{21} & q_{22} & q_{23}\\ q_{31} & q_{32} & q_{33} \end{array}\right]$$

If the scale coefficients $\lambda_{X}$, $\lambda_{Y}$, and $\lambda_{Z}$ are equal unity, then condition (11) is satisfied, the *QR* matrix is equal to the *M* matrix, the transformation is orthogonal and the transformation is a transformation by similarity. In another case, the relationship between the coordinates is described by the Equation (37), and the *QR* transforming matrix can be described as the quasi-rotation matrix, then, the transformation in question is also defined as quasi-similarity.

Stage 5:

In the fifth step, after calculating the matrix *M* (*QR*), the condition for zeroing the deviations between the coordinates of the adjustment points in the VF system (related to the centroid) and the coordinates converted to this system from the LF system using the Equations (36)/(37) is checked. The fulfillment of the condition of zero deviations L, within $1\times 10^{-4}\;m$ (ten-fold more precise than the total station measurement), can be written using the relation (39):(39)$$L=\left[\begin{array}{c} L_{X}\\ L_{Y}\\ L_{Z} \end{array}\right]=\left[\begin{array}{c} X_{iVF}\\ Y_{iVF}\\ Z_{iVF} \end{array}\right]-\left[\begin{array}{ccc} q_{11} & q_{12} & q_{13}\\ q_{21} & q_{22} & q_{23}\\ q_{31} & q_{32} & q_{33} \end{array}\right]\times \left[\begin{array}{c} X_{iLF}\\ Y_{iLF}\\ Z_{iLF} \end{array}\right]=\left[\begin{array}{c} 0.0000\\ 0.0000\\ 0.0000 \end{array}\right]$$

If the condition (39) is not met, corrections to the (quasi) *QR* rotation matrix are sought, which also includes the scale correction for each axis. Additionally, corrections to the position of the centroid of the LF system with respect to VF are sought, which can be generally written in the form of Equation (40), and after transformation in the form of Equation (41):(40)$$V=dQR\times X_{iLF}+dX_{iLF}-L$$
(41)$$\left[\begin{array}{c} V_{X}\\ V_{Y}\\ V_{Z} \end{array}\right]=\left[\begin{array}{cccccccccccc} 1 & 0 & 0 & X_{iLF} & Y_{iLF} & Z_{iLF} & 0 & 0 & 0 & 0 & 0 & 0\\ 0 & 1 & 0 & 0 & 0 & 0 & X_{iLF} & Y_{iLF} & Z_{iLF} & 0 & 0 & 0\\ 0 & 0 & 1 & 0 & 0 & 0 & 0 & 0 & 0 & X_{iLF} & Y_{iLF} & Z_{iLF} \end{array}\right]\times \left[\begin{array}{c} \begin{array}{c} \begin{array}{c} dX_{iLF}\\ dY_{iLF} \end{array}\\ \begin{array}{c} dZ_{iLF}\\ dq_{11} \end{array}\\ \begin{array}{c} dq_{12}\\ dq_{13} \end{array} \end{array}\\ \begin{array}{c} \begin{array}{c} dq_{21}\\ dq_{22} \end{array}\\ \begin{array}{c} dq_{23}\\ dq_{31} \end{array}\\ \begin{array}{c} dq_{32}\\ dq_{33} \end{array} \end{array} \end{array}\right]-\left[\begin{array}{c} L_{X}\\ L_{Y}\\ L_{Z} \end{array}\right]$$
where
V=A×X−L.

Equation (41) therefore contains the affine scale corrections with corrections to the value of revolutions (elements) and corrections to the LF system translation to the centroid—corrections of the LF system center of gravity (elements $dX_{iLF},\;dY_{iLF},\;dZ_{iLF})$. The determined corrections are added to the coordinates, and the control is the zeroing of the intercepts L in Equation (39), after taking into account corrections to the coordinates calculated by Equation (41).

Stage 6:

In the next step of transformation, back translation is performed—a shift opposite to the one that was subjected to the VF (vessel frame–fixed frame) system in the third transformation step. Reverse translation is written as the last component of the sum in Equation (23). Thus, the coordinates of the points in the LF system undergo a double change in translation. First, in the third step, it is a translation to the LF center of gravity, and in the sixth step, it is a back translation from the VF center of gravity. Thus, the transformation is completed, and its result is a point cloud in the VF system. Transformation results are assessed on common points and on check points that were common points (known in both LF and VF systems) but did not participate in determining transformation coefficients using Equations (36)/(37) and (40). In the proposed transformation model, it was assumed that the variables are uncorrelated, errors are random, and their values are small. Due to these assumptions, the normal distribution of measurements is used. As a result of the precision analysis, the mean observation error $m_{0}$, is calculated as an estimate of the variance coefficient (42), the covariance matrix *Q* (43) and finally, the mean errors of the coordinates on the connecting points (44):(42)$$m_{0}=\pm \sqrt{\frac{V^{T}\times V}{n-k}}$$
(43)$$Q={\left(A^{T}\times A\right)}^{-1}$$
(44)$$m_{X}=m_{0}\times \sqrt{Q_{XX}},\;m_{Y}=m_{0}\times \sqrt{Q_{YY}},\;m_{Z}=m_{0}\times \sqrt{Q_{ZZ}}$$
where:
n—number of matching points,k—number of observations (adjustment points) necessary to carry out the transformation (in the considered transformation model, k=3).

### 2.3. Software

For the research, the SC4W, Geonet DC (Dimensional Control) and Mathcad Professional software were used. The SC4W and Geonet DC software are designed for dimensional control, including the processing of data in the object’s own system (local coordinate system), as well as for the coordinate transformation in oblique local coordinate systems. In the SC4W and Geonet DC software, transformation by similarity with an unknown engine of calculations is used. In Mathcad Professional software, the Q-ST algorithm was implemented, and calculations were carried out. The calculations were performed with the accuracy of $10^{-4}$ meters, which for measurements in the range up to 100 m gives six significant digits. 

## 3. Results

### 3.1. Description of the Experiment

Tests on the accuracy of the Q-ST transformation were carried out based on laboratory data and on the basis of real data—collected during the measurements. Laboratory studies were based on ideal (no disturbance) and disturbed data. The disturbance to the data was generated randomly according to the normal distribution of the data. In the first step, the algorithm’s response and accuracy was verified on the test data without disturbances, then on the disturbed data, and finally, on the real measurement data. The real data were collected during the measurements on a floating ship. All data were processed using the software described in Section 2.3. Laboratory tests were performed using 4 and 5 common (adjusted) points, and field studies with 4, 5 and 6 common points. The number of control points (check points) in laboratory tests was 10. Accuracy analysis was performed independently on common and check points.

### 3.2. Experiment

A set of ideal test data has been generated and then shifted and twisted according to the transformation (1). The rotation matrix $R^{-1}$ was adopted for the rotation (the matrix *R* is described by the compound (5)), and the transformation scale was 1. The transformation parameters are presented in Table 1.

In this way, a set of laboratory data was created in two systems: VF—vessel frame (fixed frame) and LF—local frame (see Table 2 and Figure 6).

Then, the Q-ST was performed, and its results were compared with the calculations made in the software dedicated to Dimensional Control: SC4W and Geonet DC (Dimensional Control). The calculation results are summarized for two types of common points: for adjustment points and for check points (see Table 3).

The analysis of the data collected in Table 3 shows the resetting of the deviations on the combined points of adjustment and check in the Q-ST. The deviations are zero for the Q-ST, regardless of whether five fitting points (11–15 in Table 2) or four fitting points (12–15 in Table 2) were used for the transformation. In the programs for dimensional control (SC4W, Geonet DC), the deviations are not zeroed, and the errors in the position of the checkpoints after the transformation are rounded to 1 mm, despite the fact that the data used for the calculations were without disturbance.

The next step was to study the Q-ST on disturbed data. For this purpose, a new set of points was generated, which were located on a sphere with a radius of 50 m. These data were then shifted and twisted (see Table 4), and in the last step, random disturbances were added to them (see Table 5).

The following errors were adopted to generate, according to the normal distribution, pseudo-random disturbances: $m_{X}=m_{Y}=m_{z}=0.002\;m$, which gives an error in the position of the point $m_{P}=0.0035$ m. After generating pseudo-random disturbances for a set of twenty points, on a sphere with a radius of 50 m, the mean error of the point location was $m_{P}=0.0034$ m, which slightly differs from the assumed value. The generated laboratory data were therefore shifted and twisted, and then randomly generated disturbances were added to them. In this way, data was obtained in the VF and LF systems (see Table 6).

The computations on disturbed data were performed in two variants: for four adjustment points (points 12–15 in Table 2) and for five adjustment points (points 11–15 in Table 2). In the next step, for the dataset in the VF and LF systems (Table 6), the scale factor was calculated according to the formula (25), which amounted to: *λ* = 0.99999103 for four adjustment points and *λ* = 0.99997818 for five adjustment points. In all cases, the robust orthogonality criterion for calculating the scale factor was met. During the calculations with the use of the Q-ST algorithm, the condition of orthogonality of the transformation (11) for the *QR* matrix (38) was checked. The results of this verification were related to the unit matrix, which was presented for four adjustment points in Equation (45) and for five adjustment points in Equation (46):(45)$$I-QR^{T}\cdot QR=\left[\begin{array}{ccc} 0.0000154 & -0.0000203 & -0.0000114\\ -0.0000203 & -0.000036 & -0.0000203\\ -0.0000114 & -0.0000203 & -0.0000302 \end{array}\right]$$
(46)$$I-QR^{T}\cdot QR=\left[\begin{array}{ccc} -0.0000247 & -0.0000374 & 0.0000104\\ -0.0000374 & -0.0000624 & 0.0001047\\ 0.0000104 & 0.0001047 & -0.0000071 \end{array}\right]$$

The results of the Equations (45) and (46) indicate that the transformation is not orthogonal for common (adjusted) control points with a precision of six decimal places. The transformation shows elements of an affine transformation and is considered a quasi-similarity, as shown in Equation (38).

Then, the Q-ST transformation was performed, and its results were compared with the calculations made in software dedicated to Dimensional Control. The calculation results are summarized in Table 7.

The next and last stage of verification of the Q-ST was field research on real data. Measurements were made with a tilted reference frame using the total station (see Figure 7).

Measurements were made on the ship in the water, from four measuring stations. The ST1 station was the central one, which was the reference for the other measurement stations. The remaining positions were selected so that they had common points with the first position. Common points (adjustments) to the ST1 station are summarized in Table 8.

Table 9, Table 10, Table 11 and Table 12 summarize the coordinates of the points measured on individual measuring stations.

Then, the measurements from the stations ST2, ST3 and ST4 were transformed to the coordinate system of the ST1 station. The transformation was performed with the use of common points. The calculations were made in dimensional control software: SC4W, Geonet DC and with the use of the Q-ST algorithm. As a result of the calculations, the scale factor was determined from the dependence (25), which for the transformation of individual sites in relation to station 1 is presented using Equation (47):(47)$$\lambda_{21}=1.0000433\;\lambda_{31}=0.9998868\;\lambda_{41}=0.9999414$$

In all cases, the robust orthogonality criterion for scale factor calculation, checked by formula (29), was met. The orthogonality condition (11) for the *QR* matrix was also checked. The results for individual stations are presented by means of (48)–(50)—in the order ST2, ST3 and ST4 against ST1:(48)$$I-QR^{T}\cdot QR=\left[\begin{array}{ccc} -0.0001745 & 0.0004735 & 0.0065694\\ 0.0004735 & 0.0006779 & 0.0021945\\ 0.0065694 & 0.0021945 & -0.0057780 \end{array}\right]$$
(49)$$I-QR^{T}\cdot QR=\left[\begin{array}{ccc} 0.0002384 & 0.0002747 & -0.0004385\\ 0.0002747 & -0.0002029 & -0.0003900\\ -0.0004385 & -0.0003900 & 0.0005775 \end{array}\right]$$
(50)$$I-QR^{T}\cdot QR=\left[\begin{array}{ccc} 0.0007725 & -0.0003493 & -0.0015617\\ -0.0003493 & 0.0000938 & 0.0002925\\ -0.0015617 & 0.0002925 & 0.0011551 \end{array}\right]$$

The calculated scale factor (47) and the (quasi) rotation matrix indicate that the transformation is not strictly orthogonal with a six-digit precision.

The transformation errors for common (adjusted) points from individual measurement stations, in relation to the measurements from the ST1 station, are presented in Table 13. Due to the small number of common control points between the individual measuring stations, all common points were used only as the adjusted control points.

The data in Table 13 show that the smallest transformation errors are achieved using the Q-ST transformation. With five fitting points, the transformation mean errors are approximately 1.5 mm. In the other transformations—SC4W and Geonet DC—these errors are three times higher. With six adjustment points, the results achieved with the Q-ST are already close to the transformation made in the programs for dimensional control, but still smaller, this time by about 30%. In both calculation variants, with five and six points in common, SC4W and Geonet DC software give similar results and it is difficult to indicate which of them is more accurate.

The final step in verifying the Q-ST was to compare the transformation results of the measurement points (see Table 10, Table 11 and Table 12), which were not fitted (“N” designation) to transformation by similarity. The transformation of these points was performed in the following programs: PCMS (Leica), Spatial Analyzer (SA), SC4W, Geonet Dimensional Control and using the Q-ST algorithm. The discrepancy in the position of the points between individual software in relation to the Q-ST algorithm was 4.79 mm on average. This difference is about 70% of the value of the sum of errors on the common points (see Table 13) of the Q-ST compared to other methods. A sparse point cloud formed from all transformed points is shown in Figure 8.

## 4. Discussion and Conclusions

The presented method of Quasi-Similarity Transformation allows to obtain zero deviations on common points, in the case when the number of these points is four, that is, with one redundant point. The transformation of the coordinates of the connecting points was also performed in the SC4W and Geonet DC programs, which are dedicated to dimensional control, in particular in the offshore industry. Geonet DC and SC4W do not produce zero deviations on combined points when the number of match points is greater than three. Even with ideal, laboratory-generated data, errors within 1 mm are generated using these programs. In the case of five connection points, the Q-ST transformation gave three times greater precision than the software used for dimensional control purposes, which was verified, among others, on real data by transforming points measured on the ship in the inclined frame of reference. For the six connectivity points, the Q-ST gave 30% more accurate results than software used in the offshore industry. The goal of zeroing deviations on total points, set in the paper, when their number is four, has been achieved. When the number of common points is five, the deviations are no longer zero, but are still close to zero (within the range of 0.5–1.5 mm in the conducted experiments). When the number of total points is six, the precision of the Q-ST algorithm is still better, but already close to that obtained in coordinate transformation programs.

The tested Geonet DC, SC4W (for connecting points) and PCMS (Leica), Spatial Analyzer (SA), SC4W and Geonet DC software (for the remaining points) give very similar coordinate transformation results. This allows the conclusion that a similar transformation by similarity algorithm works in them, and that the subtle differences in coordinates result from the compensatory least squares approach. The Q-ST, on the other hand, has a non-standard compensatory approach—the least squares method is used separately at each stage of the calculations. The presented computational approach—the Q-ST—was verified on the following data: laboratory, disturbed and real laboratory. In each calculation variant, the Q-ST gave the smallest errors at the adjustment points, in relation to the compared methods. The assumption of the normal (Gaussian) distribution of the data and the least squares method allows us to conclude that the Q-ST method gives the most probable results of the transformation by similarity, which results directly from smaller errors in relation to other methods. Therefore, the authors believe that the Q-ST is the optimal computational method for coordinate transformations with a small number of common points (4–6 points) and can find application, in particular in the offshore industry, where it is very often necessary to perform geodetic measurements in inclined frames of reference.

## Figures and Tables

**Figure 1 sensors-20-06497-f001:**
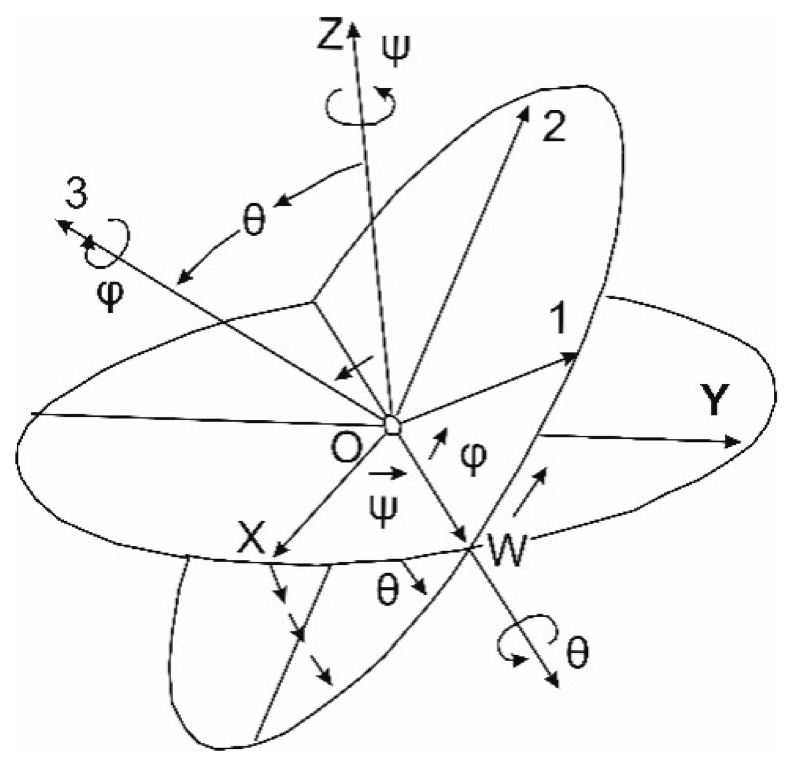
Euler’s rotation angles (shown sequence: *ψ* → *θ* → *φ*). The rotation matrix is described by the Equation (2).

**Figure 2 sensors-20-06497-f002:**
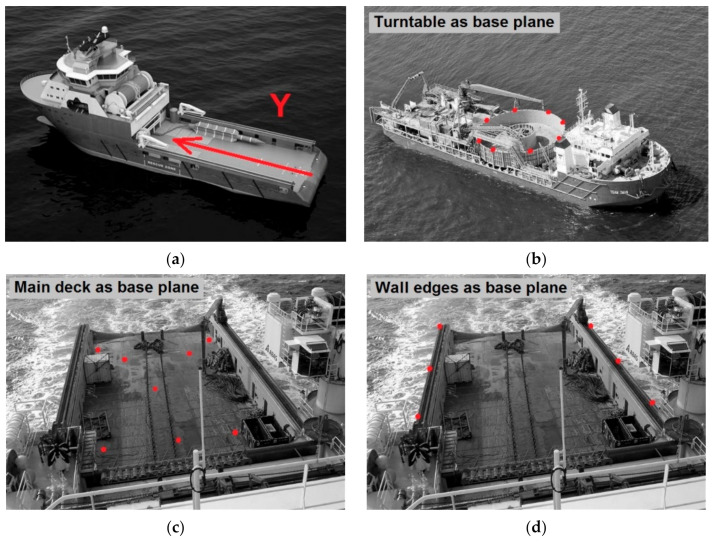
Determination of the ship’s center line (Y) and the base plane: (**a**) the ship’s center line, define the base plane based on: (**b**) turntable element, (**c**) main deck floor, (**d**) wall edges.

**Figure 3 sensors-20-06497-f003:**
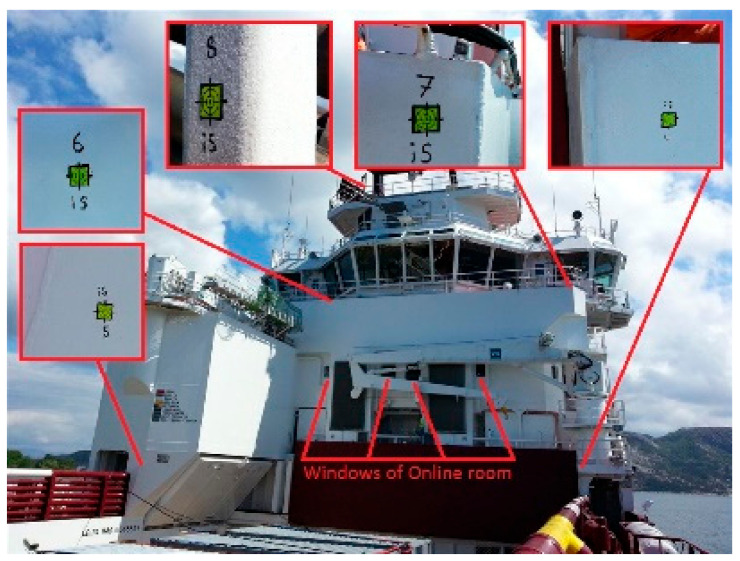
Common control points marked on the vessel.

**Figure 4 sensors-20-06497-f004:**
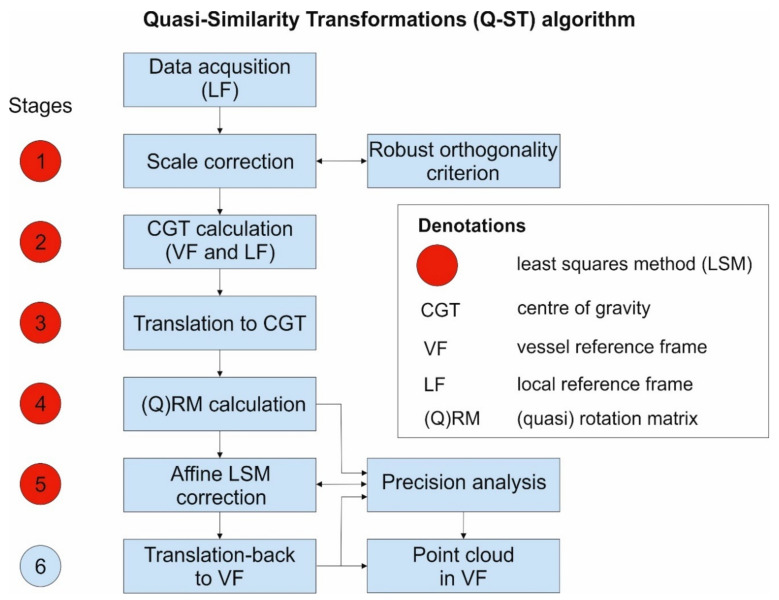
Calculation scheme of the Quasi-Similarity Transformation (Q-ST) method.

**Figure 5 sensors-20-06497-f005:**
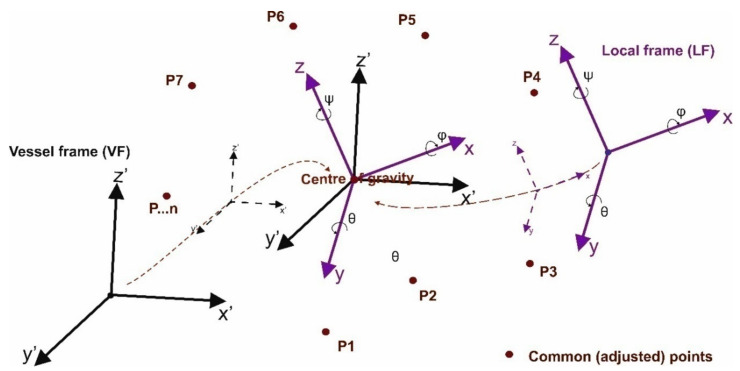
Shift of the two inclined frames of reference (VF and LF) to the centroid—center of gravity.

**Figure 6 sensors-20-06497-f006:**
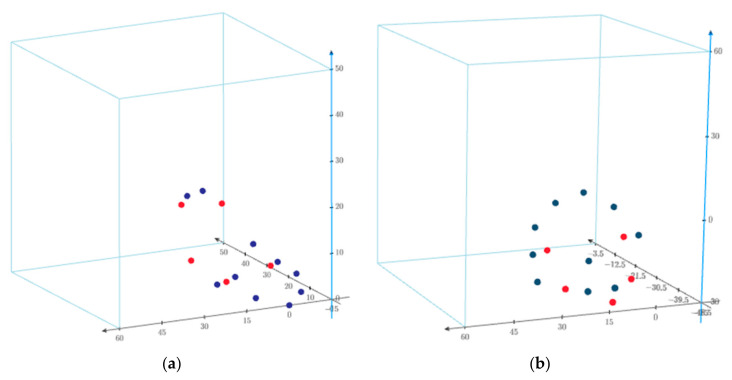
Ideal data distribution in the VF frame (**a**) and in the LF frame (**b**). Common points are marked in red color, check points are blue.

**Figure 7 sensors-20-06497-f007:**
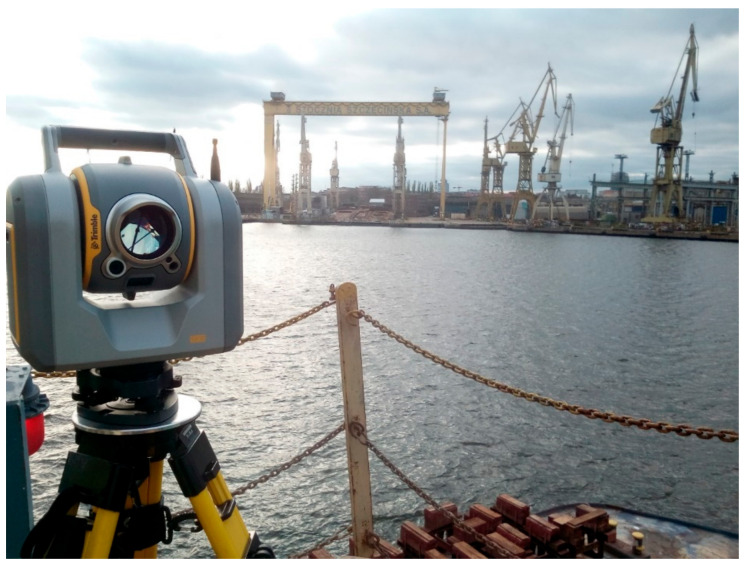
Total station measurements on a tilted reference frame (on the ship in the water).

**Figure 8 sensors-20-06497-f008:**
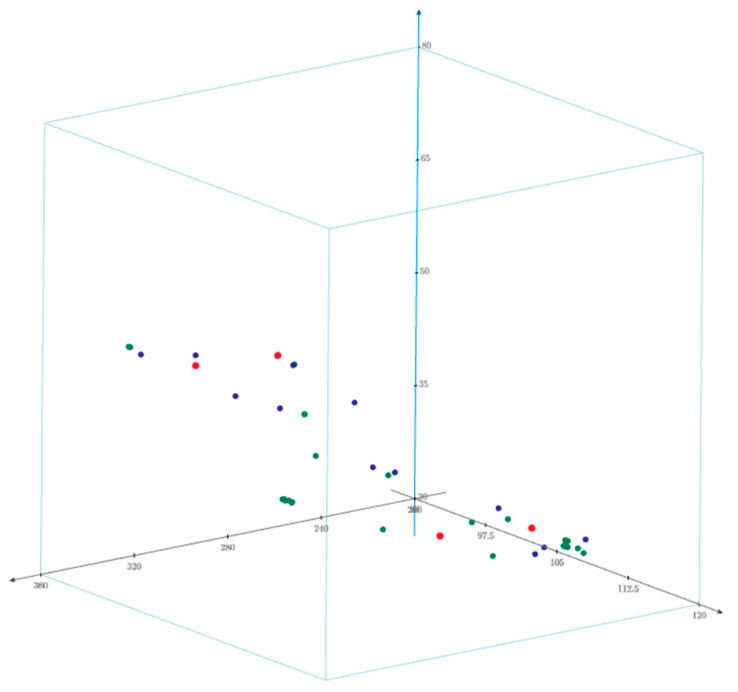
Sparse point cloud. The measuring stations are marked in red, common points are marked in blue and the measuring points are marked in green.

**Table 1 sensors-20-06497-t001:** Parameters for the transformation of the VF test dataset to the LF system.

**Value**	$\varphi \;\left({}^{\circ}\right)$	$\theta \;\left({}^{\circ}\right)$	$\psi \;\left({}^{\circ}\right)$	$X_{0}\left(m\right)$	$Y_{0}\left(m\right)$	$Z_{0}\left(m\right)$	λ
	38.57893	33.30716	227.35478	10.000	10.000	4.000	1.000000

**Table 2 sensors-20-06497-t002:** Laboratory data test set in VF and LF systems.

Point	Vessel Frame—VF			Local Frame—LF		
	*X* (m)	*Y* (m)	*Z* (m)	*X* (m)	*Y* (m)	*Z* (m)
1.	0.000	0.000	0.000	−14.006	−2.300	−3.815
2.	5.000	−8.000	1.000	−12.468	6.188	−7.763
3.	15.000	−14.000	2.000	−14.990	14.829	−15.244
4.	25.000	−15.000	2.000	−20.037	19.041	−22.846
5.	35.000	−14.000	3.500	−27.137	22.472	−29.255
6.	15.000	14.000	2.000	−32.203	−7.053	−12.257
7.	5.000	8.000	1.000	−22.304	−6.315	−6.055
8.	25.000	0.000	0.000	−28.160	6.276	−22.553
9.	40.000	0.000	15.000	−44.890	19.239	−23.995
10.	38.000	7.000	15.000	−48.060	13.083	−21.749
11.	16.000	10.000	2.000	−30.310	−3.584	−13.433
12.	23.000	−11.000	2.000	−21.364	15.229	−20.920
13.	39.000	−6.000	12.000	−38.988	22.022	−25.846
14.	42.000	6.000	12.000	−48.063	13.673	−26.814
15.	31.000	11.000	3.000	−39.967	1.302	−23.916

**Table 3 sensors-20-06497-t003:** Mean errors of transformations (standard deviation) for laboratory dataset without distortion by random errors.

Transformation	No. of Common (Adjusted) Points	Mean Errors on Common (Adjusted) Points (mm)				Mean Errors on Check Points (mm)			
		$m_{X}$	$m_{Y}$	$m_{Z}$	$m_{P}$	$m_{X}$	$m_{Y}$	$m_{Z}$	$m_{P}$
Q-ST	5	0.00	0.00	0.00	0.00 0.00	0.00	0.00	0.00	0.00
SC4W		0.33	0.10	0.18	0.38	0.42	0.49	0.70	0.96
Geonet DC		0.33	0.10	0.18	0.39	0.42	0.49	0.71	0.96
Q-ST	4	0.00	0.00	0.00	0.00 0.00	0.00	0.00	0.00	0.00
SC4W		0.33	0.14	0.20	0.41	0.44	0.52	0.72	1.00
Geonet DC		0.33	0.09	0.28	0.44	0.49	0.49	0.96	1.18

**Table 4 sensors-20-06497-t004:** Parameters for the transformation of the VF test dataset to the LF system before their disturbance.

**Value**	$\varphi \;\left({}^{\circ}\right)$	$\theta \;\left({}^{\circ}\right)$	$\psi \;\left({}^{\circ}\right)$	$X_{0}\left(m\right)$	$Y_{0}\left(m\right)$	$Z_{0}\left(m\right)$	λ
	27.35478	1.30716	1.57894	15.000	1.000	2.000	1.000000

**Table 5 sensors-20-06497-t005:** Pseudo-random disturbance for laboratory data (see Table 6).

Point	*dX* (m)	*dY* (m)	*dZ* (m)
1.	−0.0009	−0.0014	−0.0008
2.	−0.0014	−0.0010	−0.0013
3.	−0.0009	0.0011	−0.0003
4.	−0.0019	−0.0005	0.0020
5.	−0.0034	0.0002	0.0004
6.	0.0020	−0.0058	0.0009
7.	0.0017	−0.0043	−0.0010
8.	0.0018	0.0004	−0.0014
9.	0.0013	−0.0012	0.0061
10.	−0.0021	−0.0024	−0.0006
11.	0.0001	0.0002	−0.0026
12.	−0.0015	0.0015	0.0015
13.	0.0014	0.0006	−0.0008
14.	−0.0004	0.0000	−0.0024
15.	−0.0013	−0.0015	−0.0021
**Mean error—** **Calculated** **(**$m_{P}=0.0034\;m$**)**	0.0017	0.0022	0.0021

**Table 6 sensors-20-06497-t006:** Laboratory data test set with disturbances.

Point	Vessel Frame—VF			Local Frame—LF		
	*X* (m)	*X* (m)	*X* (m)	*X* (m)	*Y* (m)	*Z* (m)
1.	28.8670	28.8670	28.8680	51.967	7.250	31.873
2.	28.8670	−28.8670	28.8680	25.445	−44.025	32.681
3.	−28.8670	28.8670	28.8680	0.702	33.739	29.974
4.	−28.8670	−28.8670	28.8680	−25.821	−17.538	30.784
5.	28.8670	28.8670	−28.8680	53.281	5.661	−25.825
6.	0.0000	35.3550	35.3550	29.170	26.430	37.317
7.	0.0000	−35.3550	−35.3550	−1.700	−38.316	−32.360
8.	35.3550	35.3550	0.0000	61.370	9.242	3.146
9.	−35.3550	−35.3550	0.0000	−33.900	−21.120	1.816
10.	0.0000	−50.0000	0.0000	−9.238	−50.347	3.178
11.	50.0000	0.0000	0.0000	58.130	−28.876	4.122
12.	−50.0000	0.0000	0.0000	−30.666	17.002	0.834
13.	0.0000	0.0000	50.0000	12.594	−4.560	52.446
14.	0.0000	50.0000	0.0000	36.702	38.468	1.776
15.	0.0000	0.0000	−50.0000	14.872	−7.317	−47.492

**Table 7 sensors-20-06497-t007:** Mean errors of transformations (standard deviation) for laboratory dataset with distortion by random errors.

Transformation	No. of Common (Adjusted) Points	Mean Errors on Common (Adjusted) Points (mm)				Mean Errors on Check Points (mm)			
		$m_{X}$	$m_{Y}$	$m_{Z}$	$m_{P}$	$m_{X}$	$m_{Y}$	$m_{Z}$	$m_{P}$
Q-ST	5	0.57	0.03	0.50	0.76	2.27	3.03	2.89	4.77
SC4W		1.06	0.94	1.35	1.95	2.70	2.34	2.80	4.54
Geonet DC		1.16	0.47	1.03	1.62	2.81	2.54	2.40	4.48
Q-ST	4	0.00	0.00	0.00	0.00 0.00	2.84	2.75	2.07	4.46
SC4W		1.25	0.38	0.97	1.62	2.84	2.49	2.07	4.31
Geonet DC		1.25	0.38	0.97	1.63	2.84	2.49	2.07	4.31

**Table 8 sensors-20-06497-t008:** Points common between positions—in relation to the first position—ST1.

Point Number ST1	Common Points		
	ST2	ST3	ST4
1		+	+
2		+	+
3		+	
5	+		
6	+		
7	+		
M1		+	+
M2	+	+	+
M3	+		
M4	+		
M11			+
Suma	6	5	5

**Table 9 sensors-20-06497-t009:** Points measured at the ST1 station.

Station ST1	*X* (m) (North)	*Y* (m) (East)	*Z* (m) (Up)
Point Number			
1	234.752	116.514	29.169
2	235.198	107.448	29.096
3	260.012	100.302	32.768
5	333.558	93.845	49.364
6	337.931	104.895	49.292
7	334.844	108.814	49.302
M1	259.083	117.185	29.061
M2	299.949	101.61	49.456
M3	319.777	96.212	49.505
M4	307.113	109.849	48.809
M11	257.723	102.046	32.763

**Table 10 sensors-20-06497-t010:** Points measured at the ST2 station.

Station ST2	*X* (m) (North)	*Y* (m) (East)	*Z* (m) (Up)	Common Point (Yes—Y/No—N)
Point Number				
ST2	300.000	100.000	50.000	N
M2	264.588	99.054	48.506	Y
5	298.653	93.677	48.469	Y
6	302.242	105.017	48.403	Y
7	298.884	108.703	48.410	Y
M4	271.151	107.780	47.873	Y
FUGRO_STBD_C	273.766	102.844	56.104	N
FUGRO_PORT_C	273.770	100.120	56.141	N
FUGRO_PORT_1	273.822	100.169	56.188	N
FUGRO_PORT_2	273.844	100.131	56.189	N
FUGRO_PORT_3	273.837	100.072	56.196	N
FUGRO_PORT_4	273.813	100.049	56.187	N
FUGRO_STB_1	273.810	102.910	56.147	N
FUGRO_STB_2	273.832	102.874	56.143	N
FUGRO_STB_3	273.831	102.834	56.141	N
FUGRO_STB_4	273.810	102.797	56.144	N
GPS_PORT_1	296.980	91.978	48.604	N
GPS_PORT_2	297.054	91.978	48.602	N
GPS_PORT_3	297.119	91.914	48.599	N
GPS_PORT_4	296.945	91.956	48.604	N
GPS_PORT_5	297.022	91.984	48.603	N
GPS_PORT_6	297.125	91.882	48.599	N
GPS_STBD_1	297.020	110.743	48.519	N
GPS_STBD_2	296.966	110.711	48.519	N
GPS_STBD_3	296.888	110.722	48.518	N
GPS_STBD_4	296.925	110.709	48.518	N
GPS_STBD_5	296.995	110.722	48.519	N
GPS_STBD_6	297.032	110.762	48.518	N
M3	284.740	95.070	48.588	Y

**Table 11 sensors-20-06497-t011:** Points measured at the ST3 station.

Station ST3	*X* (m) (North)	*Y* (m) (East)	*Z* (m) (Up)	Common Point (Yes—Y/No—N)
Point Number				
ST3	300.000	100.000	50.000	N
M2	325.598	92.445	69.835	Y
M1	283.660	104.852	49.435	Y
1	259.446	102.306	49.537	Y
2	260.589	93.302	49.451	Y
3	285.881	88.085	53.128	Y
USBL_1	307.939	86.696	48.413	N
USBL_2	308.280	86.765	48.408	N
USBL_3	307.707	86.259	48.421	N
USBL_4	307.968	85.836	48.419	N
USBL_5	308.458	85.837	48.411	N
USBL_6	308.697	86.172	48.404	N

**Table 12 sensors-20-06497-t012:** Points measured at the ST4 station.

Station ST4	*X* (m) (North)	*Y* (m) (East)	*Z* (m) (Up)	Common Point (Yes—Y/No—N)
Point Number				
ST4	300.000	100.000	50.000	N
M2	356.008	90.648	69.726	Y
M1	314.558	104.564	49.317	Y
M11	313.816	89.377	53.002	Y
1	290.266	102.901	49.412	Y
2	291.082	93.860	49.329	Y
PRISM_SF	328.338	104.061	49.454	N
PRISM_SA	300.156	104.953	49.644	N
PRISM_PA	299.529	84.558	49.710	N

**Table 13 sensors-20-06497-t013:** Mean error (standard deviation) for field surveying data set.

Transformation	No. of Common (Adjusted) Points	Mean Errors on Common (Adjusted) Points (mm)				Average Errors on Common (Adjusted) Points (mm)			
		$m_{X}$	$m_{Y}$	$m_{Z}$	$m_{P}$	$m_{X}$	$m_{Y}$	$m_{Z}$	$m_{P}$
**Q-ST**	6 (ST2 to ST1)	1.86	2.34	0.75	3.09	1.44	1.82	0.55	2.39
SC4W		3.19	3.00	0.89	4.47	2.50	1.83	0.67	3.17
Geonet DC		3.24	2.88	0.92	4.43	2.54	1.92	0.77	3.28
**Q-ST**	5 (ST3 to ST1)	0.55	0.20	1.48	1.59	0.43	0.16	1.16	1.25
SC4W		3.81	1.32	2.29	4.64	3.20	1.00	1.80	4.54
Geonet DC		3.61	1.29	2.13	4.39	2.99	1.00	1.78	3.62
**Q-ST**	5 (ST4 to ST1)	1.31	0.12	0.32	1.350.00	1.03	0.10	0.26	1.07
SC4W		4.30	0.71	2.06	4.82	3.60	0.40	1.80	4.04
Geonet DC		4.56	0.80	1.93	5.02	3.80	0.52	1.68	4.19

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
