# Peer review of "Dimensioning Method of Floating Offshore Objects by Means of Quasi-Similarity Transformation with Reduced Tolerance Errors"

_sensors, 2020, doi:10.3390/s20226497_

Round 1

Reviewer 1 Report

Review for the manuscript: Sensor-958356
Dimensioning method of floating offshore objects by means of quasi-similarity transformation with zero tolerance errors

General considerations

The paper shows a workflow to obtain 3D transformation parameters using common points. The proposed approach is composed by several stages, some of them are performed as serial steps, while others using a recursive approach. I suppose that this work is an extension of the paper titled “New approach to isometric transformations in oblique local coordinate systems of reference” written and cited by the authors.

The paper is suitable for this journal. The organization of the paper could be further improved adding new section and subsection, some of them are too long.

The abstract is readable although several phrases are not so clear (see review table). The introduction provides a good description of the topic, the mathematical concepts are discussed in deep (even if some typing errors are present, see review table for further details), while the state of art is not sufficiently described. Of course, the classical approach was illustrated, but other approaches should be cited, such as proustite’s methods or principal components approach.

The motivations for this study are clear, although in the paper lacks a precise description of the use of total station in offshore environment, I strongly recommend describing the procedure adopted, inserting it into the section 2.1. In order to obtain a uniform and accurate explanation of the proposed approach an explanation of the external software employed in deep should be performed. For example, in the section 2.1

In general, the organization of the second section, where the approach followed by the authors is described, could be improved by adding a new sub-section. A discrepancy with the section 3 is detected, specifically the external software used during the experimentation should be described into the section 2.1 as well. There is only a citation of the SC4W software. The reader needs to obtain further information about the commercial software such as which is the approach employed or which is the machine precision used during the computation. Personally, I appreciated the deep description of the algorithm adopted, just minor revisions are suggested in order to avoid misunderstanding.

In this section lacks a description of the implementation of the algorithm, which programming languages was used, which precision was chosen etc.

The organization of the section 3 can be improved using the sub-section. In first sub-section (3.1) you can describe how the experimentation was conducted. I appreciated that the experimentation was conducted for step, before synthetic data (with and without perturbation) and real data. It is noteworthy the way to provide the orthogonality of the obtained rotation matrix.

The conclusions are appropriated and coherent with the paper.

The number of references is sufficient, but there are some commas which do not contain any text.

General evaluations

The contribution is readable. The organization of the paper shall be improved, while the clarity is not always present. In some parts there are insufficient details, for this to be considered an accurate description of experimentation methods especially when the authors use the total station, furthermore no details about the implementation are present.

Of course, the paper is appropriate for this journal, the quality of writing is not always satisfactory: even if in general the paper is readable. In general, the length of the paper is reasonable even if some details are omitted on the other hand several details about the rotation matrix present in section 1 can be neglected (i.e the quaternion can be cited but the relationship can be omitted because never used in the paper). There is a strong and comprehensive reference list, although some references can be added especially about alternative methods. The images are appropriate to the text and its contents. The paper is technically accurate for the description of the method, but the mathematical notation does not always seem coherent. The details in this paper are not so numerous, especially for the experimentation part where the authors seem do not know how the external software work.

Further and detailed reviews are in the table below.

Review Table

Legend:

green = suggested review ; yellow = edit is no mandatory; cyan= please edit or add that is requested

Location

Comments and review

Page 1 , Rows 11

Activity should be changed with activities

Page 1, row 12

Substitute requires with require

Page 1, row 14-15

The sentence “Each component ….” is not clear rewrite please.

Page 1, row 23

Insert a space between the two word in “SimiliratyTrasformation”

Page 1, row 25

Delete the article “a” from the sentence “Such a situation”

Page 1, row 25

Please check the word: “diuties” may be is duties

Page 1, row 28

Insert a space between the two word in “commonpoints”

Page 1, row 29

Please substitute the frames of reference with reference frame

Page 1, row 39

Please considerer to move the reference on the introduction

Page 2, row 49

Insert a space between the two word in “ComputerVision”

Page 2, row 51-52

The final part of the phrase is not so clear (from “as well as e.g. when using….”), please rewrite it

Page 3, relationship (4)

The third matrix must be changed the “phi” with “k”

Page 3,row 85

Insert a space between the two word in “intrinsicrotation”

Page 3

You can use the word “equation” in order to avoid a constant repetition of relation or relation ship

Page 4, row 113

The final part of the sentence is not very clear, please rewrite it

Page 4, relation (12)

I prefer to indicate the little angles with the symbol “dα”, although you can write it with the angle expressed in radians (but you have to specify that as you have done at the beginning of the page 5)

Page 5, relation (13)

In order to clarify the expression of the relation (13) you can say that the rotation matrix is the inverse of the matrix showed in (12)

Page 5, row 149

Please add the letter “t” before the word “ranslation”

Page 5, rows 155 and 165

Please correct “MolodenBadeks” with “Molodensky-Badekas”

Page 5, rows 166-167

The phares “centroid as the center…” should be rewritten and the first letter shall be capital.

Page 5, row 168

Please correct the word “Infintesimal”

Page 5, rows 170-171

The references are unorder, please reorder.

Page 5, rows 171-172

The sentence reports the word “equalization” I prefer to use “adjust”

Page 6, row 174

Should be more comprehensible talk about the gaussian distribution. Anyway, I prefer to indicate di function with the following function:

Page 6, equations (18) and (19)

The equations (18) and 19 could create confusion for a reader, pleas rename the L highlighting that one of them is calculated, while another is provided by the data

Page 6, equations (20-21 22)

To avoid confusion on the reader I suggest to change the XYZ (uppercase) with correspondent letter in lowercase.

Page 7, row 198

Please insert a space in the word “Quasi-SimilarityTransformations”

Page 8, row 226

Please change Figere with Figure

Page 8, row 230

Could you provide a reference that reports the correlation between geometric distribution with the accuracy

Page 8, row 239

Please define “decent distance”, please change enought with enough

Page 8, row 242

Please check and correct the word “analyse”

Page 8, row 243

Please control the word “strech”

Page 9, rows 276-277

Could be interesting to add a new figure with shows the target points in a 3D model of the ship or more easily into the General Arrangement Plan

Page 9, rows 288-289

Please insert the letter “u” and a space in the following word:” Qasi-SimilarityTransformations”

Page 9, row 291

Please insert a space into the word: “vesselframe”

Page 10, row 307

Please insert two spaces int the following word “localreferenceframe”

Page 10, row 308

Please insert two spaces int the following word “vesselreferenceframe”

Page 11, row 332

Please insert the spaces into the word: “robustorthogonalcriterion”

Page 11, rows 346

Please insert the spaces into the word: “robustorthogonalitytransformation”

Page 11, rows 348-349

The elimination (known as Data Snooping) of the observation could affect the final solution. Keep alert!

Page 11, equation (30)

Please use lowercase letter as x,y,z to avoid confusion for the reader

Page 11, row 358

Please divide the word: “commonpoints”

Page 11, row 364

I suggest to insert a further subscript during your examination such as LF or VF

Page 12, row 368 and 384

Please divide the word: “commonpoints”

Page 12, figure 5

The two translated reference system cannot provide the same point in 3D space.

Page 13, row 391

Please correct the letter in conformity with the relationship (35).

Page 13, row 392

Please insert a space in “The MatrixM”

Page 13, equation (35)

In the vector b is reported the solution, please change the first ZiVF3 (sixth row )with ZiVF2

Page 13, rows 411-415

I suggest to report the motivations about the precision of 10-4meters

Page 13, equation (3)

Report please the QR at beginning? For example: “QR=[……”

Page 14, equation (41)

Could you improve the alignment of the elements of the “A” matrix?

Page 14, row 422

Insert a point after the bracket “…elements ??iLF, ??iLF, ??iLF)”

Page 14, row432

Please divide the word: “commonpoints”

Page 14, row432

In the geodetic field the role of control points that do not participate in determining of transformation coefficient are named “Check-Points”. I prefer to use the “check points” instead of “control points”

Page 16, row 456

Insert a space after the label of the figure

Page 16, row 460

Please insert a space between “see” and Table 3

Page 16, row 466

Please insert a space between “dimensional” and “control“

Page 17, row 472

Please insert a space between “see” and Table 4

Page 17, row 473

Please insert a space between “see” and Table 5

Page 17, row 476

Please insert a space between “see” and Table 6

Page 17, row 483

Please insert a space between “see” and Table 6

Page 21 row 515

Please divide the word: “commonpoints”

Page 21 row 516

Please insert a space between “dimensional” and “control“

Page 22 row 530

Please insert a space between “dimensional” and “control“

Page 22 row 539

Please divide the word: “commonpoints”

Page 22 row 541

Please divide the word: “commonpoints”

Page 23

Please divide the word: “commonpoints” and insert a space between “dimensional” and “control“ present in the page

Conclusion

The work presented in this paper is very interesting for the marine surveys in general. In my opinion, the paper is not satisfactory but can be improved following how it was written in general considerations and evaluations as well as in the review table. In this section I would like to summarize all the required revisions:

  • Some sentence on the abstract should be rewritten
  • It is not appropriate insert reference on the abstract
  • The introduction should be completely revised, reporting other similar methodologies and approaches, furthermore some parts can be reduced as a simple citation (i.e. quaternion theory)
  • The section 2 can be improved by using further subsection and with a description of the software employed in the experimentation. Moreover, a brief description of the implementation of the presented approach should be useful.
  • In section 2 lacks a deep description of the operations on the field.
  • The section 3 can be improved by implementing the Least Square Adjustment procedure and comparing it with the presented method and together the results obtained from the external software. Even for section 3 it is strongly suggested a further division in subsection.
  • In section 3 the reader should be informed about the number of the Check Points used and where they are with a figure, furthermore, if the data snooping was been activated or not in the stage 1 (using the Chauvenet criterion) should be reported.
  • The bibliography contains several papers in the wrong format, including several comma with empty fields.
  • Use the revision table to apport minor revision.

Author Response

Dear Reviewer,

Thank you very much for the great deal of work you devoted to reviewing our manuscript. We found your comments and suggestions improving the quality of our paper.  We did our best to address and review all indicated places. Especially the following amendments and changes were made:

  1. The manuscript has been corrected according to the “review table”, included all cyan and majority of all green and yellow notes.
  2. The description of total station use in offshore industry has been added in section 2.1.
  3. Software used in research was described in section 2.3
  4. Subsections in section 3 were added, including section 3.1. Description of the experiment.
  5. References have been supplemented in the text.
  6. Minor corrections have been made to the mathematical description.

Moreover:

  1. The comments of other reviewers have been taken into account.
  2. The file with all the changes as follow up changes, has been attached in pdf format.

Once again, we would like to thank you very much for your great input into the review of our manuscript. Such a detailed review is rare. It has greatly helped us to improve our manuscript.

Authors

Reviewer 2 Report

General remarks:
- nice steps for transformation of one coordinate system into another
- a good result for a small redundant number of points, expected worse result when redundant number of points is higher
- a lot of minor faults are included into the submitted article
- references cited shall follow the rule of the Sensors (see Guide for Authors)

Author Response

Dear Reviewer,

Thank you very much for the great deal of work you devoted to reviewing our manuscript. We found your comments and suggestions improving the quality of our paper.  We did our best to address and review all indicated points to be corrected.

Moreover comments of other reviewers have been taken into account. The file with all the changes as follow up changes, has been attached in pdf format.

Once again, we would like to thank you very much for your input into the review of our manuscript. Thank you very much for your kind and positive review.

Authors

Reviewer 3 Report

The authors present a new method of quasi similarity transformation. They give a very comprehensive review of the existing approaches. They present their methodology clearly explaining every stage in detail. In addition, they present their experimental data in detail and clearly explain the procedure they followed. Finally, their conclusions are supported by their experimental results.

 I have the following comments:

In the results section can they also write the model that SC4W and Geonet DC are using, did they try to compute the transformation using a 3D affine transformation or a 3D similarity transformation?

It would be also useful to show the distribution of the control points (common points). Also it might be more appropriate to present the two datasets presented in figure 6 separately.

Author Response

(The authors gave the same response as above.)

Round 2

Reviewer 1 Report

Second review

Dear authors, thank you for your work. It is my opinion that your work was strongly improved. But there are some things to do before I give you my full acceptance.

  • Row 464 it is not clear this expression: control points (check points). Are the control or check points? Otherwise, are they both typologies? Please consider to rewrite the sentence.
  • In row 494 consider to use section instead of paragraph;
  • Row 496 it is not clear this expression: control points (check points). Are the control or check points? Otherwise, are they both typologies? Please consider to rewrite the sentence.
  • In the tables are not present the column that reports the results obtained on the check point.
  • If the data snooping was been activated or not in the stage 1 (using the Chauvenet criterion) should be reported
  • Please consider to eliminate from the title: zero tolerance errors, can be misleading. Can you re-title your work?

Author Response

Dear Reviewer,

Thank you very much once again for the great deal of work you devoted to reviewing our manuscript. Thank you for your comments and suggestions. We have put efforts to complete all of them according to your suggestions. Especially the following amendments and changes have been made (please see the answers in italic):

  1. Row 464 it is not clear this expression: control points (check points). Are the control or check points? Otherwise, are they both typologies? Please consider to rewrite the sentence – The sentence was rewritten. The term check points has been applied.
  2. In row 494 consider to use section instead of paragraph – if you do not mind, we would leave the text as it stands.
  3. Row 496 it is not clear this expression: control points (check points). Are the control or check points? Otherwise, are they both typologies? Please consider to rewrite the sentence – The sentence was rewritten. The term check points has been applied (“check points” instead of “control points” according to your suggestion).
  4. In the tables are not present the column that reports the results obtained on the check point – Yes, it is true. “Due to the small number of common adjusted control points between the individual measuring stations (in the terrain experiments), all common points were used only as the adjusted checked points” We have added this sentence to the text of the manuscript.
  5. If the data snooping was been activated or not in the stage 1 (using the Chauvenet criterion) should be reported –“ In all cases, the robust orthogonality criterion for scale factor calculation, checked by formula (29), was met”. We put this sentence into the text of the manuscript.
  6. Please consider to eliminate from the title: zero tolerance errors, can be misleading. Can you re-title your work? – We have modified the manuscript title. TO preserve the idea we propose: “Dimensioning method of floating off-shore objects by means of quasi-similarity transformation with reduced tolerance errors”

Authors